Real-time path planning for autonomous vehicle off-road driving

Ramirez-Robles Ethery
Starostenko Oleg oleg.starostenko@udlap.mx
Alarcon-Aquino Vicente
Department of Computing, Electronics, and Mechatronics, Universidad de las Américas—Puebla , Puebla , Mexico
Dennis Louise
Electronic publication date: 2024 Jul 24
Publication date: 2024
Volume: 10
Electronic Location ID: e2209
Received 2024 Feb 9; Accepted 2024 Jun 27
Copyright: ©2024 Ramirez-Robles et al.
Copyright year: 2024
Copyright holder: Ramirez-Robles et al.
License: This is an open access article distributed under the terms of the Creative Commons Attribution License, which permits unrestricted use, distribution, reproduction and adaptation in any medium and for any purpose provided that it is properly attributed. For attribution, the original author(s), title, publication source (PeerJ Computer Science) and either DOI or URL of the article must be cited.
License URL: https://creativecommons.org/licenses/by/4.0/

Keywords: Machine vision, Semantic image segmentation, Off-road autonomous driving, Navigation path planning

Funding: CONACYT (National Council for Science and Technology) The authors received no funding for this work. Ethery Ramirez-Robles received a scholarship for Doctorate studies by CONACYT (National Council for Science and Technology) in Mexico. The funders had no role in study design, data collection and analysis, decision to publish, or preparation of the manuscript.

==============================
Background

Autonomous driving is a growing research area that brings benefits in science, economy, and society. Although there are several studies in this area, currently there is no a fully autonomous vehicle, particularly, for off-road navigation. Autonomous vehicle (AV) navigation is a complex process based on application of multiple technologies and algorithms for data acquisition, management and understanding. Particularly, a self-driving assistance system supports key functionalities such as sensing and terrain perception, real time vehicle mapping and localization, path prediction and actuation, communication and safety measures, among others.

Methods

In this work, an original approach for vehicle autonomous driving in off-road environments that combines semantic segmentation of video frames and subsequent real-time route planning is proposed. To check the relevance of the proposal, a modular framework for assistive driving in off-road scenarios oriented to resource-constrained devices has been designed. In the scene perception module, a deep neural network is used to segment Red-Green-Blue (RGB) images obtained from camera. The second traversability module fuses Light Detection And Ranging (LiDAR) point clouds with the results of segmentation to create a binary occupancy grid map to provide scene understanding during autonomous navigation. Finally, the last module, based on the Rapidly-exploring Random Tree (RRT) algorithm, predicts a path. The Freiburg Forest Dataset (FFD) and RELLIS-3D dataset were used to assess the performance of the proposed approach. The theoretical contributions of this article consist of the original approach for image semantic segmentation fitted to off-road driving scenarios, as well as adapting the shortest route searching A* and RRT algorithms to AV path planning.

Results

The reported results are very promising and show several advantages compared to previously reported solutions. The segmentation precision achieves 85.9% for FFD and 79.5% for RELLIS-3D including the most frequent semantic classes. While compared to other approaches, the proposed approach is faster regarding computational time for path planning.

Introduction

Autonomous vehicle (AV) navigation is quite new and still an open problem; the proposed solutions promise many benefits for the economy and society. Currently, two principal scenarios for autonomous driving are examined: on-road and off-road driving. The on-road scenario is for urban cities, when the lane markings, defined cues, speed signs, and pavement roads among other features are taken into account. Conversely, there are uneven surfaces in the off-road driving, where there are no explicit delimiters; so, vegetation, obstacles, and different terrains must be analyzed in real time for path prediction. Additionally, off-road terrains are more prone to changes due to different weather conditions, for example, there would be mud instead of dry soil and puddles can appear after the rain. In several countries, including the USA and Mexico, at least one-third of the roads are unpaved. Therefore, although there are multiple approaches for off-road vehicle navigation that in general use synthetic scene datasets, it is a challenge to improve their performance for autonomous driving assistance in a real-world unstructured environment.

The research methodologies used for supporting AV navigation are subdivided into modular and end-to-end approaches. In the modular approach, there exist some subtasks to be solved independently such as AV localization, camera perception, position mapping, path planning, vehicle control, and others. In the end-to-end approach, AV assistance system looks like a black box, where the inputs are the data received from sensors and the outputs are low-level commands for the vehicle driving control. In this research, we focus on a modular approach to propose an assistive real time system oriented to work on resource-constrained devices.

The principal module that defines the quality of off-road AV assistance is terrain perception and scene understanding, which are based on precise image segmentation and objects classification. One of the most promising technique is semantic segmentation used in many applications, for instance, for detecting brain tumors (Kumar, Negi & Singh, 2019; Myronenko & Hatamizadeh, 2020), recognition of road signs in urban environments (Timbus, Miclea & Lemnaru, 2018; Khan, Adhami & Bhuiyan, 2009), processing satellite images (Wurm et al., 2019; Saifi & Singla, 2020), detection of plant diseases in agriculture (Singh & Misra, 2017; Milioto, Lottes & Stachniss, 2018), and autonomous driving (Kaymak & Uçar, 2019; Treml et al., 2016), among others. It is important to highlight that during autonomous driving the resolution, perspective, and visual angle of cameras vary for different vehicles; therefore, the segmentation approach must be adjusted to particular environment. Usually, one or several sensors mounted on the AV are used for video stream gathering, such as video cameras, radiolocation system RADARs, LiDARS, GPS, and inertial measurement units (IMUs). In some cases, there is a preprocessing step that combines the data from the sensors; thus, the inputs to scene perception module are not just raw readings. Current approaches also use deep neural network (DNN) for scene perception that provide noticeable enhancing in the segmentation accuracy (Valada et al., 2017; Maturana et al., 2018). However, the suitability of DNN due to its computational cost derived from many sequential layers with tensorial operations must be evaluated, particularly, for AV driving assistance systems with limited resources and real-time computing needs.

When the information about scene is obtained, the AV path planning and route prediction is accomplished. To solve this task, usually the global and local planning are provided. While in the former the goal is to find the best route considering the environment map, in the latter a vehicle follows the global route and considers ambient changes to avoid obstacles and predict a collision free path. Several approaches for path planning are based on application of neural networks (Elshall et al., 2020; Jafri & Kala, 2022) however, there are others that exploit modified versions of Dijkstras algorithm (Brooks et al., 2022; Szczepanski & Tarczewski, 2021), A* algorithm (Erke et al., 2020; Hong et al., 2021), state lattices (Kushleyev & Likhachev, 2009), curve interpolation planners, probabilistic road maps (PRM) and Rapidly-exploring Random Trees (RRT) (Nieto et al., 2010), among others.

To assess the proposed approaches, the modular framework for assistive real-time driving in off-road scenarios oriented to resource-constrained devices has been designed and tested, which is considered as a practical contribution of this research, while theoretical contributions consist of an original approach for image semantic segmentation fitted to off-road driving scenarios and adapting the shortest route searching A* and RRT algorithms to AV path planning. Additionally, the prominent models for image segmentation in the state-of-the-art, which are UpNet (Valada et al., 2017), Dark-fcn (Maturana et al., 2018), and DeepLab (Chen et al., 2018a; Chen et al., 2018b) in combination with MobileNet-v2 (Sandler et al., 2018) have been exploited in the proposed approach. The base techniques for development of path planning were RRT and A* algorithms that combine data from images and point clouds.

The rest of this article is organized as follows. ‘Materials & Methods’ includes a description of well-known approaches and used datasets. Then, ‘The Proposed Approach’ describes the proposed approach. The experimental results are discussed in ‘Results and Discussion’. Finally, ‘Conclusions’ presents the concluding remarks.

Materials & Methods

In this section, the evaluation of some neural network architectures to be used in the proposed approach for semantic segmentation is presented. The description of the proposed algorithm for AV path planning is also discussed.

UpNet is a convolutional neural network (CNN) for semantic segmentation of unimodal and multimodal/multispectral images in unstructured forested environments. UpNet refers to encoder–decoder architecture and uses two main components: contraction and expansion. The contraction structure employs a very deep convolutional network VGG 13-layer model (Simonyan & Zisserman, 2015) generating a low resolution mask, while the expansion part refines the segmentation masks. That is, the contractive side uses convolution layers followed by a pooling layer, and the expansive side is composed of one upsampling layer plus a rectified linear unit (ReLU) followed by a convolution layer. Finally, the authors proposed to add a spatial dropout after the first and last refinement layers.

In Maturana et al. (2018), the authors propose two different CNNs for semantic segmentation comparing their results with the Freiburg Forest benchmark. The first CNNs-FCN (fully-connected network) is based on VGG-CNNs, and second Dark-FCN employs the Darknet architecture. Both networks exploit convolution and pooling layers, upsampling deconvolution layers, and element wise sum layers. Additionally, CNNs-FCN uses the normalization layer. The conducted experiments show that the Dark-FCN is faster than the CNNs-FCN and has a higher resolution, and provides more precise segmentation results. Similarly to Valada et al. (2017), the authors in Maturana et al. (2018) use two modalities: RGB images from a camera and a point cloud from a LiDAR sensor. This approach first performs the image segmentation based on color features and then involves the mask alongside the point cloud as inputs for the semantic mapping. The output of the system is what they call 2.5-D semantic map. This approach was tested on a real AV and it was capable to successfully navigate through off-road environments. However, the Dark-FCN showed better results than the approach reported in (Valada et al., 2017) to identify roads and sky.

DeepLab (Chen et al., 2018a; Chen et al., 2018b) is a model for semantic segmentation mainly used for object segmentation like persons, vehicles, animals. The version v3+ implements a spatial pyramid pooling module and novel encoder–decoder structure. The encoder captures contextual information by pooling features at different resolutions, whereas the decoder obtain sharp object boundaries. Several network backbones such as Xception65 (Chollet, 2017), Xception71, MobileNet-v2 (Sandler et al., 2018), and MobileNet-v3 (Howard et al., 2019) are recommended to use as image feature extractors.

After analyzing the most promising approaches for image segmentation, the DeepLab has been selected as a base model. However, in the proposed hybrid approach for semantic image segmentation, the DeepLab model has been modified using atrous convolution, also known as dilated convolutions. The atrous convolution provides multiple convolutions on extracted information and delivers a wider field of view at the same computational cost. In our assessment, MobileNet-v2 architecture is selected since it is a fast network used in mobile low resource-constrained devices. It uses depth-wise and point-wise separable convolutions to reduce model size and complexity. It is important to note that there are available several pre-trained checkpoints for use in DeepLab with new datasets. Some of checkpoints are pre-trained on different datasets like ADE20K (Zou et al., 2014; Zhou et al., 2017) Cityscapes (Cordts et al., 2016), MS-COCO (Lin et al., 2014), ImageNet (Deng et al., 2009), and PASCAL-VOC (Everingham et al., 2010).

Regarding the well-known shorted path search algorithm, the most outstanding is A* proposed by Hart (Hart, Nilsson & Raphael, 1968). A* is based on the Dijkstra’s algorithm, in which a route is searched between the start and the goal positions by examining in each iteration the neighbors of a parent node. However, in A* the selection of new nodes considers the path cost; therefore, the nodes with the lowest cost are selected. Another one used for path planning is Rapidly-exploring Random Trees (RRT) (Kuffner & LaValle, 2000). The algorithm creates a tree from the initial point avoiding collisions and stopping once the tree reaches the goal point. Compared to other algorithms, RRT presents a faster solution; nevertheless, the created path is not necessarily optimal. To solve the problem of optimality the modified RRT* has been created, when the new nodes are selected, it also considers the cost of the path. Our proposed algorithm based on the concepts of the best path planning approaches is discussed in the next section.

For assessment of the systems for AV driving assistance, several open datasets have been considered. One benchmark dataset is Freiburg Forest Dataset (FFD) (Valada et al., 2017) that contains different scenes oriented to off-road AV navigation. FFD consists of multispectral data obtained from two cameras, a Bumblebee2 stereo vision and a modified dashcam, recording both at 20 Hz. It contains 336 images with their respective ground truth labels and a set of 15,000 raw images. FFD includes six semantic classes: obstacle, trail, sky, grass, vegetation, and void. The selected images for training and testing present some variability in lightning conditions; also, some records include the presence of snow.

The FFD is divided into 230 training images and 136 validation images. It also contains manually annotated pixel-wise ground truth segmentation masks. FFD is smaller than other datasets; however, the majority of other datasets found in the literature are more general and oriented to urban scenes, whereas FFD is specifically oriented toward off-road tasks. FFD contains not only RGB images but also Depth and Near-infrared data. The dataset also include a multispectral channel fusion called NRG, which is an image with Near-infrared, Red, and Green in each channel. Due to the fact that there is significant presence of vegetation in off-road scenes, two indexes related to vegetation NDVI and EVI are included to dataset. The Normalized Difference Vegetation Index (NDVI) estimates the quantity, quality, and vegetation development. The Enhanced Vegetation Index (EVI) enhances the vegetation signal with improved sensitivity in high biomass regions.

Another open source of scenes for AV off-road navigation testing is multimodal RELLIS-3D dataset oriented for scene understanding (RELLIS-3D dataset: v1.0 release, 2021). This dataset includes RGB camera images, LiDAR point clouds, stereo images, GPS measurement, and IMU data. Compared to FFD, it contains a larger amount of annotated data with 13,556 LiDAR point clouds, 6,235 images, and 20 semantic classes. Nevertheless, RELLIS-3D is an imbalanced dataset, the more common classes found are grass, tree, bush, and sky. A company named Appen labeled the classes and then data were assigned to trained annotators. However, in the segmentation mask, the annotations have differences from one image to another even if the RGB images are almost identical. RELLIS-3D provides a training (available 3,302 images), validation (1,672), and testing splits (983).

The Proposed Approach

The proposed hybrid image segmentation approach and AV path-planning algorithm are implemented in a modular framework for AV assistive driving in off-road environments as shown in Fig. 1.

Figure 1 Block diagram of the proposed vehicle assistive system for autonomous driving.

Sources: https://github.com/unmannedlab/RELLIS-3D, CC BY-NC-SA 3.0; http://deepscene.cs.uni-freiburg.de/#datasets, Copyright Autonomous Intelligent Systems. All rights reserved.

The data are retrieved from a camera and LiDAR that feed three modules: perception, traversability, and path planning. The perception module performs semantic segmentation over RGB images by generation a segmentation mask. The traversability module creates a binary occupancy grid map based on segmentation mask and the objects height information from the LiDAR point clouds. Finally, the path-planning module is implemented to predict the area that the vehicle needs to follow to arrive at a certain goal.

Perception module

To implement the perception module, the transfer learning to retrain modified DeepLab (called modDeepLab) is used, while MobileNet-v2 as the network backbone is exploited to perform the feature extraction. The principal operations of the traversability module are shown in Fig. 2.

Figure 2 Methodology for creation of the perception module.

The network is programmed to learn scenes in off-road environments. To train the network three datasets have been employed, particularly, FFD with 330 images, FFD_Aug (described in next section) with 990 images, and RELLIS-3D with 3,302 images. In order to decrease the processing time, the input images were resized to 300 × 300 pixels. A momentum optimizer of 0.9 for the network with a base learning rate of 0.0001 for 100,000 steps have been used.

Two pre-trained checkpoints were implemented and evaluated. The first one, referred in this article to as ADE20K, is based on ADE20K and ImageNet datasets, while the second one named as MS-COCO is based on ImageNet, MS-COCO, and VOC 2012 (Microsoft COCO Dataset, 2017). The selection of the two pre-trained checkpoints is defined by the content of the datasets, particularly, when the images of forested environments that contain trees, sand, and ground are required. For the ADE20K checkpoint, the atrous spatial pyramid pooling is not used, it is applied only during the decoding. For the validation stage of modDeepLab, 136 images were used for the FFD and FFD_Aug, while 1,672 images were taken for the RELLIS-3D. In the semantic segmentation, the most used performance metric is the Jaccard index, also known as Intersection over Union (IoU). This metric measures the similarity between the ground truth mask and the predicted mask. It is defined as the number of pixels that overlaps between the ground truth mask and the prediction mask divided by the total number of pixels of both masks. The performance metrics used to compare the reported results with commonly known works were mean IoU (mIoU) shown in Eq. (1), and Frequency Weighted IoU (FWIoU)—in Eq. (2):

(1) IoU=1C∑CTPcTPc+FPc+FNc,

(2) FWIoU= ∑CPcPTPcTPc+FPc+FNc,

where C represents the number of classes and c ∈ [1...C], TPc stands for true positives, FPc false positives, FNc false negatives, Pc all positives and P represents the total number of pixels.

Therefore, the Eq. (1) represents the mean IoU of the used semantic classes, while Eq. (2) describes the average IoU of classes, weighted by the number of pixels in the class. For example, applying the proposed segmentation method to image with the semantic class person, the distribution of TPc, FNc, and FPc can be presented in visual form. Figure 3 illustrates how the IoU measures the overlap between the ground truth mask and the predicted mask. The TPc, shown in pink, correspond to the correctly identified pixels, where the prediction intersects with the ground truth. The purple region represents the FNc that are the pixels missed by the model but present in the ground truth; while the FPc depicted in yellow are the incorrectly classified pixels as person.

Figure 3 Visual interpretation of true positive (TPc), false positive (FPc), and false negative (FNc) for semantic class person.

Traversability module

The principal operations of the traversability module are shown in Fig. 4. In this module, the segmentation mask from the perception module with the point clouds from the LiDAR are fused to create a binary occupancy grid map. Since data comes from two different sensors, the first task is to transform the point clouds coordinates into the camera system coordinates. Therefore, the pinhole camera model is used to transform 3D coordinates from the LiDAR to 2D coordinates in a camera system taking into account the distortion coefficients as well as the intrinsic and extrinsic parameters of the camera (Laible et al., 2012). The intrinsic parameters are unique and inherent to a given camera; these describe the internal characteristics of the camera, such as its focal length, distortion, among others. The extrinsic parameters define the position and orientation relative to the world coordinate system and are captured by the rotation matrix and the translation vector. It is essential to consider both sets of parameters to map accurately the 3D points onto 2D images.

First, with the intrinsic parameter the following matrix is created (see Eq. 3): (3) A=fx0x00fyy0001,

where fx and fy stands for the pixel focal length, and x 0 and y 0 indicates the principal point (all values are expressed in pixels). Then with the extrinsic parameter, a new matrix according to Eq. (4) is created: (4) B=R00t01,

where R is 3 × 3 rotation matrix and t is translation vector. Next, by multiplying the matrices A and B we complete the accurate mapping from 3D points of the LiDAR sensor system to the image plane. With this new set of coordinates, it is possible to check, which points are in the same Field Of View (FOV) of a camera. Using the focal length and the sensor size, we obtain the horizontal (HFOV) and vertical (VFOV) fields of view of each point using Eq. (5) (5) HFOV=2⋅arctanHx2⋅Fx,VFOV=2⋅arctanHy2⋅Fy,

where Hx and Hy represent the width and height of the image, respectively. All points are compared in the x and y axes against the HFOV and VFOV, and only if both angles are within the FOV, the coordinates from that point are saved. In real cameras there exist some distortions generated by lenses so, the distortion coefficients to project the previous obtained coordinates into the real 2D points in pixel coordinates can be obtained, for example, using projectPoints procedure from OpenCV. With the final 2D pixel coordinates, z-value array is created and it is used to store the measurement on the vertical axis corresponding to the height of the detected points along with their coordinates in the camera system. Figure 5 shows an example of the 3D points visualization into the 2D images after being transformed onto the camera image plane.

Figure 4 Internal processes of the traversability module.

Figure 5 3D LiDAR points mapped onto camera image with semantic class obstacle.

Source: https://github.com/unmannedlab/RELLIS-3D, CC BY-NC-SA 3.0.

It is important to note that camera sensors capture visual details such as texture, color, and shape; however, they can be susceptible to shadows, glare, and poor visibility. On the other hand, LiDAR sensors are less affected by lighting conditions and provide more precise depth measurements of the surrounding environment in day and night. By fusing data from camera and LiDAR the system has more comprehensive understanding of the environment. Therefore, a camera presents visual data for object detection and terrain segmentation, while LiDAR provides depth information from objects and the terrain. Figure 6 presents the comparison of the binary occupancy maps generated from the image obtained using RGB camera only, and RGB camera and LiDAR. There are clear differences in the resulting occupancy maps, when using only RGB camera images; the inaccuracies in the segmentation model may lead to the erroneous representation of open space when there are trees and branches. In autonomous driving, the safety of passengers and pedestrians is crucial and such inaccuracies could represent a potential safety hazard.

Figure 6 Comparison of binary occupancy maps: (A) generated by RGB camera; (B) generated by RGB camera and LiDAR; (C) original image with mapped 3D LiDAR points.

Source: https://github.com/unmannedlab/RELLIS-3D, CC BY-NC-SA 3.0.

Finally, a binary occupancy grid map with the same size of the original RGB image is generated taking also into account dimensions of the vehicle. To assign the values of the grid map and to define space available for AV passing, two criteria are applied: first, if the pixel belongs to a non-traversable class such as tree, pole, sky, vehicle, person, fence, barrier and rubble, and second, if the pixel belongs to an obstacle. Obstacles are considered as any point detected with a height equal to or greater than 10cm from the ground. This matrix feeds the next AV path-planning module.

Path-planning module

To create this module, an evaluation of two general techniques was first carried out to find the most suitable for specific tasks of AV traversable path-planning. Figure 7 shows a set of procedures for assessment and adapting A* and RRT algorithms for generating a traversable route in off-road environment. The module has three inputs, which are the binary occupancy grid map, the start, and end coordinates of route.

Figure 7 Methodology for the creation of the path-planning module.

A* algorithm finds an optimal path with the fewest possible number of nodes by evaluating the minimum cost that is equal to the calculation of a heuristic function (Hart, Nilsson & Raphael, 1968). Generally, the heuristic function used in grids is the Manhattan distance, when a movement is allowed in four directions only. For a movement in eight directions the Chebyshev distance is applied. In order to adapt A* algorithm to path planning for off-road navigation, the Euclidean distance (see Eq. 6) as heuristic function h is proposed to use for the movement in any direction. (6) h=current nodex−goal nodex2+current nodey−goal nodey2.

Another RRT optimal route searching algorithm (Kuffner & LaValle, 2000), which is frequently used in several scientific reports, has been exploited for assessment of the proposed approach. The challenge was to adapt it to AV path planning using as input the same binary occupancy grid map from the traversability module. The RRT algorithm finds a feasible path by creating a tree from the initial node checking if an object exists to avoid collisions. In each iteration, the tree is expanded through space generating a new random node. This node is connected to the nearest available node after checking if no obstacle between these two nodes are detected taking into account the occupancy grid map. The pseudocode of the proposed algorithm for path searching is shown below.

Initialize: maxDistance, maxIterations, counter = 0, tree = [start]

node _new ← start

while counter <maxIterations do

while distance (node _new, goal) >d _threshold do

node _target = randomNode()

node _nearest = tree.nearestNeighbor(node _target)

node _new = extend(node _nearest, node _target, maxDistance)

if obstacleFree(node _nearest, node_new) then

tree.add(node_new)

end if

end while

resulting_path ← tree.Traceback(node _new)

end while

if distance(node _new, goal) <error then return tree

else print Error: goal not reached

end if

Compared to other algorithms, RRT presents a faster solution and takes into account the presence of obstacles to avoid; however, the route created will not necessarily be optimal and depending on the maximum number of defined iterations, it may not reach the solution.

Results and Discussion

In order to assess the performance of the proposed approach, several tests have been conducted using a specially designed modular framework for assistive driving in off-road scenarios oriented to resource-constrained devices.

Image segmentation results and discussion

For the perception module, as mentioned in the previous section, MobileNet-v2 was selected as the backbone of modDeepLab with two different checkpoints ADE20K and MS-COCO used to retrain the network on base of two datasets FFD and RELLIS-3D. Additionally, in order to grow more the original FFD dataset, the images were augmented using Imgaug (Jung et al., 2016) by exploiting Python library created to augment images for machine learning projects. It contains different techniques, e.g., perspective transformations, contrast changes, cropping, blurring, among others. Four augmenters have been selected and applied. The first image set has been created using coarse dropout that deletes 5% of all pixels in some channels and assigns them different random values. Since no geometric transformations were applied, masks remain the same. For the second set of augmented images Gaussian blur, Perspective transform, and Flip transformations were applied in a random order for each image. The perspective transformation was performed on a random scale between 0.01 and 0.15, and the image size was not affected. The 50% of the images were flipped vertically and 50% horizontally. Each image was blurred with Gaussian kernel with a sigma between 0.0 and 3.0. The metrics to evaluate in each class were mean Intersection over Union (mIoU) and Frequency Weighted Intersection over Union (FWIoU) calculated according to Eqs. (1) and (2), which are based on the confusion matrix shown in Fig. 8. All the training and evaluation were run on a Laptop with a Core i7-8750H and an NVIDIA GTX 1050Ti GPU.

Figure 8 Confusion matrix with result obtained for: (A) ADE20K and B) MS_COCO checkpoints, respectively.

As shown in Table 1, the results for both ADE20K and MS-COCO checkpoints were very similar; however, the best performance was with ADE20K. It was suspected to see an increase in the weighted IoU, when dataset augmentation was performed. While there was a slight improvement, the results did not change that much.

Table 1 Semantic segmentation results for FFD, FFD_Aug, and RELLIS-3D datasets.

Dataset	Pretrained checkpoint	Image size (pixels)	mIoU	FWIoU	
FFD	ADE20K	300×300	72.98%	84.64%	
FFD_Aug			72.97%	84.73%	
RELLIS-3D			34.90%	79.49%	
FFD	MS_COCO	300×300	70.47%	83.51%	
FFD_Aug			72.67%	84.60%	
RELLIS-3D			31.29%	80.50%	

In the case of the augmented FFD_Aug, the vegetation and sky classes obtained the best segmented results. In contrast, the object class had the lowest percentage of detection. Meanwhile for the RELLIS-3D pole, water, building, log and fence were the classes with the lowest detection. It is known that neural networks need many examples to learn, and the classes that had the lowest result on both datasets were the least present. In addition, the class object on FFD is a general class that contains different objects like pedestrians, people riding bicycles, logs, benches, buildings at the horizon. Consequently, it was difficult for the network to learn features from a very diverse class.

It is important to mention that for off-road navigation some types of obstacle can be omitted due to rare appearance of buildings, people riding bicycles, logs, etc. After assessment of the proposed approach with the most frequent semantic classes of obstacles, the best mIoU for 300 × 300 size images increases to 73.42%, while the FWIoU achieves 85.9%, both on FFD_Aug dataset. For comparison, for image size 1024 × 768 the achieved mIoU was 75.28%, while FWIoU reaches 86.25%.

In addition to the confusion matrix and the computed mIoU and FWIoU metrics, which are benchmark performance metrics, the global accuracy also has been evaluated. This metric measures the ratio of correctly classified pixels to the total number of pixels, regardless the class. The results with ADE20K checkpoint were 87.31%, while the MS_COCO checkpoint achieved 88.40%. It is important to note that the global accuracy may be influenced by imbalances in class distribution like sky and grass that predominate in the dataset.

Figure 9 shows a visual comparison of the results for experiments with ADE20K and MS-COCO checkpoints of modDeepLab. Each row corresponds to the indicated dataset. The first column shows the original RGB image, which is the input to the network. The second column depicts the ground truth mask, which is the expected result. The third and fourth columns are results with ADE20K and MS-COCO, respectively.

Figure 9 Visual comparison of the segmentation results for ADE20K and MS-COCO checkpoints.

Sources: https://github.com/unmannedlab/RELLIS-3D (CC BY-NC-SA 3.0) and http://deepscene.cs.uni-freiburg.de/#datasets (Copyright Autonomous Intelligent Systems. All rights reserved.)

One of the main differences, which can be observed in vegetation such as trees and shrubs, is the level of details. Predictions using both checkpoints is not as detailed in the area of leaves and branches as seen in the ground truth. It also can be noted that, when there are more classes as in RELLIS-3D, the module fails to classify some small and thin objects like fences mistaking them as trees or people.

Some training images were taken with the sun giving directly to the camera. In these images, there are some sunrays, in which the camera captured as white pixels. The modDeepLab was not affected by those white pixels. It detected and classified the majority of the images correctly. However, the presence of shadows affected the results. In some cases, the network classified soil pixels as grass, and in a small number of images, they were classified as vegetation. The same happened with puddles despite being a specified class in RELLIS-3D, the network misclassified several pixels depending on the reflex in the puddle.

There are several metrics that can be used for performance assessment of semantic segmentation approaches such as F1 score that evaluates the harmonic mean of the precision and recall (Jebamikyous & Kashef, 2021) or success rate that represent the percentage of correctly classified pixels in input image compared to the ground truth (Papadeas et al., 2021). However, mIoU and FWIoU are still common benchmarks currently used elsewhere (Muhammad et al., 2022). The obtained results from modDeepLab assessment are comparable to other works, which also use mIoU and FWIoU metrics as shown in Table 2.

Table 2 Comparison of the semantic segmentation results of modDeepLab with other approaches.

Approach	Image size	mIoU	FWIoU	
UpNet	300×300	76.68%	85.30%	
Dark-FCN	300×300	60.35%	89.41%	
modDeepLab	300×300	73.42%	85.90%	

It is important to conclude that the proposed approach has similar performance overcoming slightly the Dark-FCN (Maturana et al., 2018) in mean IoU, and UpNet (Valada et al., 2017) in FWIoU. The best result, particularly, for identifying roads and the sky was from Dark-FCN using FWIoU of 89.41% as well as it segments one image during 35 ms (Maturana et al., 2018). Nevertheless, the modDeepLab turned out to be faster taking only 32 ms per image. In addition, Dark-FCN had IoU for class object about 5.03%, while modDeepLab achieves up to 34.42% for the same class, noting also that the result of UpNet with 45.31% was better.

Path-planning results and discussion

For the path-planning module the modified A* and RRT algorithms have been tested and compared by using two metrics: the pixel path cost and the time that algorithms take to predict a new path. These metrics are the most common for assessment of path-planning algorithms among others such as the shortest path length that refers to the total distance traveled by the vehicle, the safety that include minimum distance to obstacles (Chen et al., 2017), the collision rate, and the path smoothness (Chu, Lee & Sunwoo, 2012).

Table 3 presents results of 100 running with each of ten random images. The first row shows the image number, the second and third rows present the path costs in pixels and the fourth and fifth - the execution time in seconds for A* and RRT algorithms, respectively.

Table 3 Comparison of the path cost and execution time in seconds of A* and RRT algorithms.

		1	2	3	4	5	6	7	8	9	10	
Path cost	A*	565.98	607.99	314.14	501.84	1,765.3	723.85	903.14	1,864.3	241.01	1,815.3	
RRT	690.63	741.57	356.8	624.78	2,044.1	861.63	1,071.2	2,117.9	311.92	2,142.6	
Execution time	A*	0.812	0.817	0.714	0.797	1.711	0.922	0.995	1.751	0.784	1.772	
RRT	0.096	0.141	0.371	0.053	0.236	0.073	0.090	0.177	0.041	0.527	

In AV driving assistance, the time is crucial, since there are usually several processes running at the same time corresponding to each module so that, a vehicle can take its decisions in real time. As a result, after observing the results with both algorithms, the RRT has been selected as the most appropriate option for the proposed approach. Although RRT has a higher path cost in all executions, it was faster with the total average time around 0.2329 s against 1.2568 s achieved by A* algorithm.

In Fig. 10 some examples of predicted paths obtained by using RRT algorithm are depicted. It can be seen that based on specifications of terrain classes, RRT generates a traversable route with slightly higher cost. However, it avoids detected by occupancy grid obstacles and provides about one second faster the path planning compared to A* algorithm.

Figure 10 Predicted routes obtained by RRT algorithm plotted on the original RGB image.

Source: https://github.com/unmannedlab/RELLIS-3D, CC BY-NC-SA 3.0.

Implementation of real-time Semantic Segmentation app for mobile devices

To assess whether a real-time terrain perception can be carried out on resource-constrained devices from different platforms, Semantic Segmentation application (SSapp) has been designed and tested using Huawei P30 Pro and Samsung S10+ cellphones on Android 10 and iPad Pro 11 with iOS 15.4.1. The modDeepLab runs with TensorFlow format nonetheless, smartphones cannot run the entire TensorFlow library. Instead, there is a version called TensorFlow Lite (TFLite) developed by Google to test its own segmentation models. After arranging the format of data used in the proposed approach to TFLite format, smartphones can run semantic segmentation.

Figure 11 shows the SSapp’s interface for Android-based cellphones that visualizes the preprocessing time, the model execution time, mask flatten time, full execution time, and labels with their randomly designated color. In Fig. 12 an interface of the designed SSapp for iPad Pro is also depicted.

Figure 11 Screenshot of the Semantic Segmentation application for Android-based cellphones with default information.

Figure 12 Screenshot of designed Semantic Segmentation application with example of segmentation mask obtained on iPad Pro.

The user is capable of activate or disable the use of GPU to run the segmentation. In addition, the captured image and the segmentation mask are presented on a screen. The assessed app runs in average 274 ms for full scene segmentation that includes preprocessing, model inference, post processing, and visualization of 300×300 images on the Android devices, while on iPad in average 460 ms are required.

In order to test the semantic segmentation app “on the wild”, several videos were recorded on two different roads with Huawei P30 Pro smartphone at 60 fps with a resolution of 1,920 × 1,080. Videos were taken on sunny days in different hours and one of the recordings was taken at sunset. The smartphone was put in two different positions. The first position was inside the car next to the rearview mirror and the second position was outside the car at the sunroof. Comparing with various researches, one of the main differences between real scenes with the scenes from FFD is the vegetation color (Otsu et al., 2016; Brooks & Iagnemma, 2012). The vegetation is greener in the videos, whereas in FFD, there are more trees with only branches and brown vegetation. Another notable difference is the presence of tree shadows on the road and several puddles (Chen et al., 2018a; Chen et al., 2018b; Rosique et al., 2019). In the experiments with smartphone inside the car, reflexes from the windshield were considerably affecting the overall segmentation, as well as the shadows from the tree on the ground also affected the segmentation results with both ADE20K and MS-COCO. In the MS-COCO checkpoint results, puddles were mainly classified as objects, whereas in the ADE20K, results were not affected. Not surprisingly, the app was able to segment the video recorded at sunset, considering it was only trained with those scenarios. For all the experiments in sunset, puddles were also classified as objects. Nevertheless, the app was still capable of correct segmenting sky and soil if compared to the visual results in the day videos.

Conclusions

In this article, a new approach for assistive driving in off-road scenarios is presented. The obtained results are very promising and show several advantages compared to the solutions presented in scientific papers. Due to the fact that an improvement of image processing with modDeepLab in the perception module, it is possible to obtain competitive results in the segmentation mask generation for the most of semantic classes used in AV navigation in off-road environments.

Particularly, after assessment of the proposed approach with the most frequent semantic classes of obstacles, the best mIoU for 300×300 size images increases to 73.42%, while the FWIoU achieves 85.9%, both on FFD_Aug dataset. For comparison, for the image size 1024x768 the mIoU is 75.28%, while FWIoW reaches 86.25%.

In addition, the proposed perception module generates semantic segmentation mask with more than 8% shorter time compared to the Dark-FCN model (Maturana et al., 2018) for processing images from the same FDD and RELLIS-3D datasets. Finally, the real-time path prediction is feasible by using a modified RRT algorithm that achieves route search with five times less compared to the frequently used A* algorithm. For example, although RRT has a higher path cost in all executions, it is faster with the total average time around 0.2329 s against 1.2568 s achieved by A* algorithm.

The practical contribution of this article consisted in the development of a framework that is used for assessment of the proposed approach and testing it either on records from common datasets or on real “wild” scenarios.

It is important to mention that the used LiDAR sensor and camera are susceptible to heavy rain, snow or fog generating considerable errors in scene understanding (Royo & Ballesta-Garcia, 2019). Therefore, the acceptable functionality of the proposed perception module considers only good weather conditions. The results of this research also show that it is possible to use images with a lower resolution and still obtain competitive results. Despite the fact that our results have been compared to other works, there are some semantic classes with a lower detection rate. Further studies should use more balanced datasets with a higher amount of examples for each class; therefore, the network can identify better features and improve the results (Badrinarayanan, Kendall & Cipolla, 2017). This work supports the idea that it is possible to use a camera from a smartphone and run the segmentation on the same device as a more accessible and economical option for driving assistance. In addition, it is necessary to carry out other experiments with more powerful hardware to make a deeper analysis on the effect of training with different parameters and architectures. Finally, we believe there still exist a big gap between researchers on urban scenarios and in off-road. There exists a need for more off-road datasets so, researchers can continue to study and improve the state-of-the-art of AV navigation in this type of environment.

Supplemental Information

Supplemental Information 1 Semantic segmentation of video frames

Supplemental Information 2 Computing time and cost oF A* and RRT path planning algorithms

Additional Information and Declarations

Competing Interests

Author Contributions

Data Availability

Vicente Alarcon-Aquino is an Academic Editor for PeerJ

Ethery Ramirez-Robles conceived and designed the experiments, performed the experiments, analyzed the data, performed the computation work, prepared figures and/or tables, and approved the final draft.

Oleg Starostenko analyzed the data, prepared figures and/or tables, authored or reviewed drafts of the article, and approved the final draft.

Vicente Alarcon-Aquino performed the experiments, authored or reviewed drafts of the article, and approved the final draft.

The following information was supplied regarding data availability:

The ADE20K dataset is available at: https://groups.csail.mit.edu/vision/datasets/ADE20K.

The Microsoft COCO dataset is available at: https://public.roboflow.com/object-detection/microsoft-coco-subset.

The Freiburg Forest dataset is available at: http://deepscene.cs.uni-freiburg.de.

The RELLIS-3D dataset is available at GitHub: https://github.com/unmannedlab/RELLIS-3D.

The codes are available in the Supplemental Files.

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
