# Peer review of "Real-time path planning for autonomous vehicle off-road driving"

_PeerJ Computer Science, doi:10.7717/peerj-cs.2209_

## Round 0.1 · original submission · Minor Revisions

Reviewer 2 has a number of minor corrections that should be made specifically that
- The first letters of words containing the expansions of abbreviations must be capitalized. For exp., Autonomous Vehicle (AV).
and
- It may be better if the terms TP, TN, FP and FN, which are the terms denoting metrics in Eq.1 and Eq. 2, are visualized with a diagram.

Reviewer 1 requests some more major revisions mostly to establish performance improvements generated by
- integration of camera and LIDAR
- a broader comparison with state-of-the-art methods in off-road navigation
Even if a comprehensive comparison can't be performed this should be touched on and the reasons why these comparisons can't be made.

Reviewer 1 also requests more clarity around the alignment between LIDAR and camera that should be provided.

**Language Note:** PeerJ staff have identified that the English language needs to be improved. When you prepare your next revision, please either (i) have a colleague who is proficient in English and familiar with the subject matter review your manuscript, or (ii) contact a professional editing service to review your manuscript. PeerJ can provide language editing services - you can contact us at [email protected] for pricing (be sure to provide your manuscript number and title). – PeerJ Staff

Reviewer 1 ·

Basic reporting

The language of the paper is suitable for its technical content, but there is room for improvement in clarity and consistency. The introduction and background effectively contextualize the work within the broader autonomous vehicle (AV) navigation field. The article maintains a professional structure, and the tables and figures are clear. Additionally, the paper presents a diverse set of results, encompassing segmentation accuracy metrics and path planning efficiency measures.

One strength of the proposed approach is the real-time testing on different roads and lighting conditions using mobile devices.

Experimental design

The research question in the paper appears to be well-defined, relevant, and meaningful within the context of autonomous vehicle (AV) navigation, mainly focusing on off-road scenarios. The paper presents a modular approach for assistive driving in off-road environments. The perception module utilizes a modified DeepLab Model to perform semantic segmentation on RGB images obtained from the camera, while the traversability module fuses LiDAR point clouds with the segmentation results to create a binary occupancy grid map for scene understanding. The path planning module utilizes the occupancy grid to predict a path for the vehicle.

The paper needs a more explicit demonstration of the advantages of integrating camera and LiDAR data for performance enhancement. Showcasing specific examples or case studies where the combined sensor fusion leads to superior performance compared to using either sensor alone would strengthen the paper's argument for integration.

The proposed approach is compared with the existing works in terms of segmentation accuracy and path planning efficiency. However, a more comprehensive comparison with state-of-the-art methods in off-road navigation could provide better insights into the performance of the proposed approach. Including metrics such as success rate and comparing against a broader range of benchmarks would strengthen the evaluation of the proposed methodology.

In traversability mode, The FOV calculation can determine whether Lidar points fall within the camera's view. However, for the spatial alignment between Lidar and camera data, Lidar orientation and rotation with respect to the camera are also required. Is this not important for the generation of the binary occupancy grid?

Further to my previous comment, the visualization of mapping between the Lidar and image points will provide more clarity to the readers.

Validity of the findings

no comment

Cite this review as

·

Basic reporting

• In this paper, a novel approach including semantic image segmentation and route planning is proposed for autonomous driving of vehicles in off-road environments. The results obtained show advantages when compared to the results in the literature.
• The paper addresses a topic that is interesting, up to date, and in the scope of the journal.
• The paper is satisfactory organized.
• The proposed methodology is remarkable and novel. In addition, the main contribution of the paper is well highlighted.
• The title is applicable and appropriate.
• The figures and tables are placed correctly.
• The references are accurately cited.
• The results are promising.
• It may be better if the terms TP, TN, FP and FN, which are the terms denoting metrics in Eq.1 and Eq. 2, are visualized with a diagram.
• The paper needs a few minor changes as listed below:
- The first letters of words containing the expansions of abbreviations must be capitalized. For exp., Autonomous Vehicle (AV).
- The “Conclusions” section should be expanded and made more meaningful by using numerical data in the experimental results.

Experimental design

no comment

Validity of the findings

no comment

Additional comments

• I recommend that the paper be accepted after above revisions.

Cite this review as

---

## Round 0.2 · Minor Revisions

Reviewer 1 has asked for one additional minor correction - ensuring legibility of the title of figure 6C. As soon as this is fixed, I hope to be able to accept and won't be sending out for further review.

Reviewer 1 ·

Basic reporting

The authors have improved the manuscript. With these improvements, the revised paper can now be accepted for publication with a minor adjustment: ensuring that the title of Figure 6C is legible.

Experimental design

no comments

Validity of the findings

no comments

Additional comments

no comments

Cite this review as

·

Basic reporting

I believe that the changes made are appropriate for the acceptance of the paper.

Experimental design

no comment

Validity of the findings

no comment

Additional comments

no comment

Cite this review as

---

## Round 0.3 · Minor Revisions

I still can't read the title on 6C - the caption is fine, but the lettering above image C is illegible even when zoomed in.

---

## Round 0.4 · accepted · Accept

The caption for Figure 6C is now legible and the manuscript is ready for publication.